# Data Pruning: Counting the Frequency of Loss Transition from Above-Average to Below-Average (FATB) During Early Training

## Abstract

In this paper, we propose a novel data pruning algorithm named FATB, which aims to remove potentially redundant data and inherent noise in the original dataset during model training, thereby identifying a core subset of data rich in informational value necessary for effective model training. To quantify the informational content of individual samples during training, we introduce a method that compares the loss value of each sample with the average loss of all samples, resulting in four fundamental scenarios. These scenarios can be combined to describe the transition process of a sample loss relative to the average loss throughout the training. The informational value of a sample is then derived based on the influence weights associated with these four scenarios. However, computing the influence weights for all four scenarios requires substantial computational resources. To address this challenge, we approximate the sample transition process using a core scenario, where the loss of a single sample transitions from above the average loss to below it between adjacent phases of model training, to estimate the informational value of the sample. Additionally, since the informational contribution of a single sample may vary across different phases of training, we employ an early stopping iteration to determine the count of such core transitions, thereby obtaining a core subset of data enriched with high informational value for model training. Extensive experimental results demonstrate that the proposed method effectively eliminates redundant and noisy data, significantly enhances model performance when training on smaller target-scale core subsets, and remains effective on large-scale datasets.

## 1 Introduction

In recent years, with the continuous advancement of deep learning, large-scale datasets, which contain richer information, have been widely used in model training to achieve better performance and enable more practical applications. However, acquiring such datasets requires higher storage costs, and training on them demands greater computational resources, which are often prohibitively expensive for many companies or institutions. Research on large datasets has revealed that they often contain redundant data, noisy data, and other types of data that contribute minimally or even negatively to model training. Therefore, developing suitable data pruning algorithms for large-scale datasets can help remove such data, reduce model training time, lower storage and computational costs, and maintain or only slightly compromise model performance.

According to Kaplan et al. (2020), the relationship between model performance and dataset size follows a power law, indicating that simply following larger datasets is not optimal. Instead, a balance must be struck between model performance and dataset size, as datasets beyond a certain size offer diminishing returns. Thus, identifying the core subset of a large dataset is crucial to effectively enhance model performance. Subsequently, Sorscher et al. (2022) demonstrated that data pruning can overcome the power-law scaling limitations of test error with respect to dataset size in neural networks. Using a teacher-student perceptron model, they proved that a well-designed

pruning metric can achieve exponential scaling, suggesting that dataset size can be further reduced without significant loss in model performance.

Existing data pruning algorithms can be categorized into three types based on their core parameters: zero-order algorithms (Toneva et al. (2018);Zhang et al. (2024);Xia et al. (2022);He et al. (2024)), first-order algorithms (Paul et al. (2021);Tan et al. (2023);Maharana et al. (2023);Koh et al. (2019);Zheng et al. (2022);Coleman et al. (2019)), and second-order algorithms (Yang et al. (2022);Tan et al. (2025);Lin et al. (2024);Yang et al. (2024);Ben-Baruch et al. (2024);Chen et al. (2024)). Algorithms later in this sequence generally require more time and computational resources. Current research has largely focused on zero-order methods because of their efficiency and competitive performance. This paper proposes a novel zero-order algorithm that requires less time and computational resources while outperforming existing approaches. The algorithm reformulates the data pruning problem by considering four basic scenarios derived from comparing the loss values of individual samples with the average loss of all samples between two consecutive training iterations. A key scenario among these is identified to efficiently solve the pruning problem. By evaluating the information content of the samples based on this scenario, a core subset of data is selected. Since the method relies only on the loss values and their averages from two adjacent iterations-without requiring long-term loss or gradient computations-it significantly reduces time and resource demands.

Beyond high costs, current data pruning methods face additional challenges: Huang et al. (2024) considers individual sample loss and its variation but ignores inter-sample correlations; Toneva et al. (2018) and Paul et al. (2021) perform well at low pruning ratios but suffer sharp performance degradation at high ratios, even underperforming random pruning; Tan et al. (2023) accounts for correlations between samples but overlooks sample-specific dynamics during training; Zhang et al. (2024) captures phase-wise loss changes but ignores trends. To address these issues, the proposed algorithm compares individual sample losses with the global average loss, incorporating sample correlations. It also uses an early stopping strategy to mitigate performance drop at high pruning ratios and leverages the key scenario to capture sample-specific dynamics and trends.

In summary, the main contributions of this paper are as follows:

- We propose a novel data pruning algorithm (FATB) that reframes the pruning problem using four basic scenarios, identifies a core scenario among them, and applies it to efficiently prune datasets.

- The algorithm only requires comparing loss variations between consecutive iterations and does not involve computationally expensive operations such as gradient calculations, significantly reducing time cost. By terminating proxy model training earlier—following the trend that lower pruning ratios correspond to later optimal phases—it further cuts training time and enhances model performance at high pruning ratios.

- The method effectively removes redundant and noisy data across datasets with different distributions, generalizes well to various model types, and scales efficiently to large datasets, substantially reducing training time while improving model performance.

## 2 RELATED WORKS

Data pruning research currently focuses on two main categories: static data pruning (also known as coreset selection) (Toneva et al. (2018);Paul et al. (2021);Tan et al. (2023);Zhang et al. (2024);Xia et al. (2022);Yang et al. (2022);Tan et al. (2025);He et al. (2024);Maharana et al. (2023);Marion et al. (2023);Park et al. (2023)) and dynamic data pruning (Qin et al. (2023);Huang et al. (2024);Raju et al. (2021);Hong et al. (2024)). Static data pruning involves first training a proxy model, then using metrics obtained from the training process as algorithm parameters to assign scores to data points. Samples with low scores are removed to form a new data subset. Dynamic data pruning uses metrics obtained from earlier stages of model training as algorithm parameters to score data points. Low-scoring samples are pruned, and the remaining samples continue training. This process forms one cycle, which is repeated until the entire training process is complete. Whether sample scores are recalculated in each cycle depends on the specific algorithm and scenario.

**Static Data Pruning:** Toneva et al. (2018) defined forgetting events and observed that certain samples are frequently forgotten while others are rarely forgotten. They proposed that removing the

unforgettable samples can enhance training efficiency without compromising the model's generalization capability. Paul et al. (2021) introduced two data pruning methods based on GraNd score and EL2N score. The GraNd score measures a sample's expected impact on training based on the norm of its gradient, facilitating the early identification of important samples. The EL2N score approximates the GraNd score more efficiently using the norm of the error between the predicted probability and the true label, enabling effective pruning at high ratios. Tan et al. (2023) rapidly estimates the MoSo score based on the alignment between a sample's gradient and the average gradient. This method achieves high-ratio pruning with linear complexity by evaluating a sample's importance through its effect on the optimal empirical risk, using a first-order approximation of gradient consistency to avoid the high cost of leave-one-out retraining. TDDS (Zhang et al. (2024)) employs a dual-depth strategy—an inner loop assessing sample contribution and an outer loop focusing on generalizability—to dynamically integrate sample importance throughout training, significantly improving post-pruning model performance.

**Dynamic Data Pruning:** InfoBatch (Qin et al. (2023)) is an unbiased and lossless training acceleration framework based on dynamic pruning. In each iteration, it performs a random soft-pruning on samples with losses below the average, according to a preset probability. The gradients of retained low-loss samples are scaled to approximate the original gradient expectation, while the full dataset is used in later stages to reduce variance. Huang et al. (2024) introduced the DynImpt algorithm, a dynamic data selection method. After a brief warm-up phase to address cold-start, it builds a window of recent loss values for each sample. Importance is dynamically evaluated from three dimensions: the current loss value, the instability of recent importance (discrepancy), and the inconsistency between individual and average importance trends. These normalized components are equally weighted into a composite score, based on which a fixed proportion of samples near the median within each class are selected for the next training iteration.

## 3 PROPOSED METHODS

### 3.1 PRELIMINARIES

In this paper, we denote the complete training dataset as $S = \{(\boldsymbol{x}_n, \boldsymbol{y}_n)\}_{n=1}^N$, where $\boldsymbol{x}_n \in \mathbb{R}^D$ and $\boldsymbol{y}_n \in \mathbb{R}^{1 \times C}$ are samples independently and identically distributed (i.i.d.) drawn from a natural distribution $\mathcal{D}$. The neural network model parameterized by weight matrix $\boldsymbol{\theta}$ is denoted as $f_{\boldsymbol{\theta}}$. Based on these definitions, the optimization objective of $f_{\boldsymbol{\theta}}$ on $S$ is to minimize the empirical risk

$$L(S; \boldsymbol{\theta}) = \frac{1}{N} \sum_{n=1}^N \ell(f_{\boldsymbol{\theta}}(\boldsymbol{x}_n), \boldsymbol{y}_n),$$

where $f_{\boldsymbol{\theta}}(\boldsymbol{x}_n) \in \mathbb{R}^{1 \times C}$ outputs the predicted probabilities for each class.

Data pruning aims to construct a core data subset $U = \{(\boldsymbol{x}_m, \boldsymbol{y}_m)\}_{m=1}^M$, where $U \subset S$, such that $U$ can achieve comparable performance to $S$. Defining the pruning ratio as pr, the size of the core data subset $U$ after pruning is $M = N \times (1 - \text{pr})$. The objective of data pruning can be formulated as:

$$\mathbb{E}_{\substack{(\boldsymbol{x}, \boldsymbol{y}) \sim \mathcal{D} \\ \boldsymbol{\theta}_0 \sim P_{\boldsymbol{\theta}_0}}} \left[ \ell \left( f_{(U, \boldsymbol{\theta}_0)}(\boldsymbol{x}), \boldsymbol{y} \right) \right] \simeq \mathbb{E}_{\substack{(\boldsymbol{x}, \boldsymbol{y}) \sim \mathcal{D} \\ \boldsymbol{\theta}_0 \sim P_{\boldsymbol{\theta}_0}}} \left[ \ell \left( f_{(S, \boldsymbol{\theta}_0)}(\boldsymbol{x}), \boldsymbol{y} \right) \right], \tag{1}$$

where $f_{(U, \boldsymbol{\theta}_0)}$ and $f_{(S, \boldsymbol{\theta}_0)}$ represent the network models obtained by training on $U$ and $S$, respectively, using weights $\boldsymbol{\theta}_0$ initialized from distribution $P_{\boldsymbol{\theta}_0}$.

### 3.2 METHOD

This paper primarily considers the scenarios involving the change in individual sample loss values and the change in the mean loss of all samples, which can be categorized into four basic scenarios (as shown in Figure 1).

The variation patterns throughout the entire training process can be composed of combinations of these four basic scenarios. Instead of spending significant time determining the optimal combination of these four basic scenarios, we can identify the most influential basic scenario among them to serve as the core basic scenario, which approximates the optimal combination.

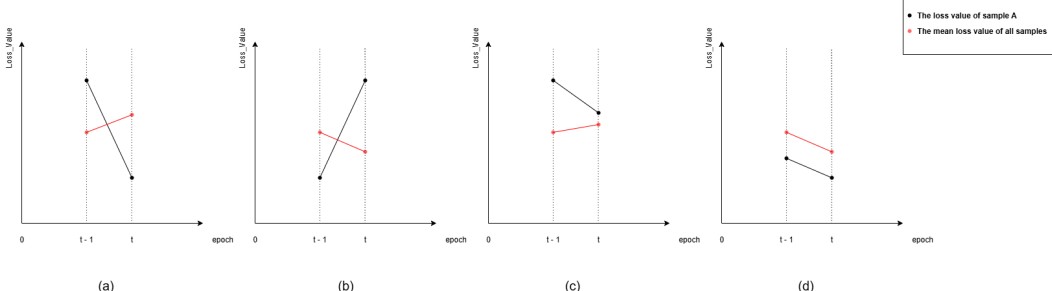

Figure 1: During the model training process, the loss value of an individual sample in two consecutive epochs can exhibit four distinct change patterns relative to the mean loss of the entire dataset: (a) The loss value of Sample A shifts from being greater than the mean loss of the entire dataset in the previous epoch to being less than or equal to the mean loss in the current epoch. (b) The loss value of Sample A shifts from being less than or equal to the mean loss of the entire dataset in the previous epoch to being greater than the mean loss in the current epoch. (c) The loss value of Sample A remains consistently greater than the mean loss of the entire dataset across both consecutive epochs. (d) The loss value of Sample A remains consistently less than or equal to the mean loss of the entire dataset across both consecutive epochs. It is important to note that the mean loss of the entire dataset is dynamic and may either decrease or increase from the previous epoch.

This paper adopts an approach similar to Toneva et al. (2018), determining the information content of samples based on the statistical occurrence of the target scenario. Simultaneously, an effective measure for high pruning ratios, namely using a cutoff iteration number, is employed, similar to Zhang et al. (2024). The difference from Zhang et al. (2024) lies in the fact that the results from all early stages of the proxy model training are beneficial for our method and do not need to be discarded. The specific methodological formulas are as follows:

The average loss over the entire dataset can be expressed as:

$$L(S; \theta_t) = \frac{1}{N} \sum_{n=1}^{N} \ell(f_{\theta_t}(x_n), y_n). \tag{2}$$

The individual sample loss scenario 1 (hard-to-learn) can be expressed as:

$$\text{HTL}((x_n, y_n); t) = \mathbb{1}\left(\ell(f_{\theta_t}(x_n), y_n) > L(S; \theta_t)\right). \tag{3}$$

The individual sample loss scenario 2 (easy-to-learn) can be expressed as:

$$\text{ETL}((x_n, y_n); t) = \mathbb{1}\left(\ell(f_{\theta_t}(x_n), y_n) \leq L(S; \theta_t)\right). \tag{4}$$

Basic scenario (a) can be expressed as:

$$\text{BSA} = \mathbb{1}\left(\text{HTL}((x_n, y_n); t-1) \wedge \text{ETL}((x_n, y_n); t)\right). \tag{5}$$

Basic scenario (b) can be expressed as:

$$\text{BSB} = \mathbb{1}\left(\text{ETL}((x_n, y_n); t-1) \wedge \text{HTL}((x_n, y_n); t)\right). \tag{6}$$

Basic scenario (c) can be expressed as:

$$\text{BSD} = \mathbb{1}\left(\text{HTL}((x_n, y_n); t-1) \wedge \text{HTL}((x_n, y_n); t)\right). \tag{7}$$

Basic scenario (d) can be expressed as:

$$\text{BSC} = \mathbb{1}\left(\text{ETL}((x_n, y_n); t-1) \wedge \text{ETL}((x_n, y_n); t)\right). \tag{8}$$

The target problem can be expressed as:

$$\text{TP}((x_n, y_n); t) = \mathbb{1}(\text{BS}), \tag{9}$$

where $t \geq 0$, $1 \leq n \leq N$, and BS represents one of BSA, BSB, BSC, and BSD.

Building upon prior theoretical foundations established in Swayamdipta et al. (2020); He et al. (2024), we initially conjectured that basic scenarios (c) and (d) would demonstrate inferior performance characteristics, while either scenario (a) or (b) would likely emerge as the dominant pattern. This intuition stemmed from the theoretical framework suggesting that transitions between loss states provide more informative signals than persistent states.

Our empirical investigation, detailed in Section 4.3 and summarized in table 5, substantiates this hypothesis and specifically identifies basic scenario (a) as the core mechanism underlying the observed phenomena. This scenario captures the critical transition where samples move from above-average to below-average loss between consecutive training iterations.

To streamline our analytical framework and enhance computational efficiency, we subsequently restrict our attention to basic scenario (a). This simplification yields the following formal representation:

$$\text{STP}((x_n, y_n); t) = \mathbb{1}(\text{BSA}), \tag{10}$$

where STP denotes the simplified target problem focusing exclusively on the transition from above-average to below-average loss states.

In summary, we propose a zero-order data pruning algorithm (see Appendix B.1) that counts the number of times, prior to a cutoff iteration, a sample transitions from being hard-to-learn to easy-to-learn (as defined in Equation 10) across consecutive training iterations. To facilitate understanding of the algorithm's operation during proxy model training, a simple example is provided: simulating the loss value changes for two samples, A and B, alongside the change in the mean loss of all samples (including A and B). Our algorithm is then applied to count the points along the loss change curves that satisfy the defined basic scenario. The final intermediate results from a complete proxy model training session are illustrated in Figure 2.

## 4 EXPERIMENTS

### 4.1 EXPERIMENTAL SETTINGS

**Datasets and Models:** To evaluate the effectiveness of the new algorithm FATB, three popular datasets were employed: CIFAR-10 (Krizhevsky et al. (2009)), CIFAR-100 (Krizhevsky et al. (2009)), and ImageNet-1K (Deng et al. (2009)). Two model architectures, namely ResNet-18 (He et al. (2016)) and ResNet-50 (He et al. (2016)), were used for the primary evaluations. Furthermore, to verify the generalization capability of FATB across different models, two additional architectures, VGG-16 (Simonyan & Zisserman (2014)) and ShuffleNet-v2 (Ma et al. (2018)), were also utilized.

**Baselines:** FATB was compared against the following data pruning algorithms: Random pruning: Each sample in the entire dataset has an equal probability of being selected; a number of samples equal to n * pr (total samples * pruning ratio) are randomly removed. Forgetting (Toneva et al. (2018)): The number of times an example is incorrectly predicted after being correctly predicted in a previous epoch. EL2N (Paul et al. (2021)): The L2-norm of the error vector. MoSo (Tan et al. (2025)): Determines sample importance by assessing the change in the optimal empirical risk when a specific sample is removed. TDDS (Zhang et al. (2024)): Employs a dual-depth strategy to balance the integration of training dynamics with the identification of representative samples. The inner level estimates the individual contribution of each sample during training by projecting its gradient onto the accumulated gradient, while the outer level focuses on the variability of these contributions to highlight samples with good generalizability.

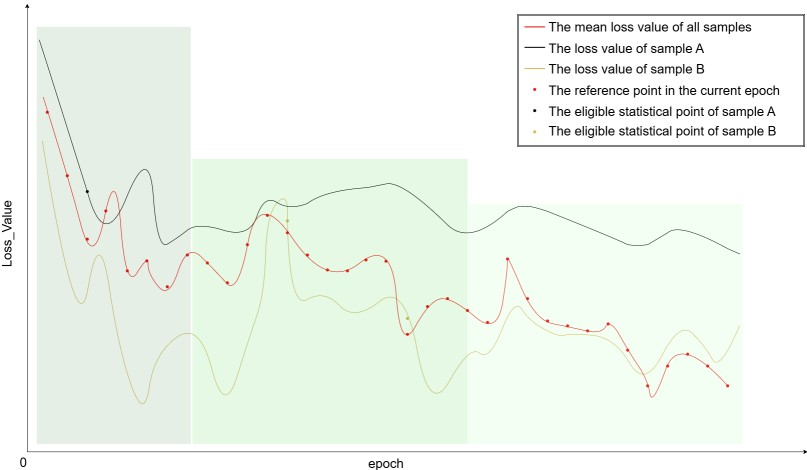

Figure 2: It illustrates the loss value trajectories of two samples, A and B, during the training of the surrogate model. The three rectangular boxes demarcate three distinct training phases. Observing from left to right: in the first phase, sample A is marked once; in the second phase, sample B is marked twice; and in the third phase, neither sample receives a mark. The selection priority of FATB is contingent upon the chosen cutoff iteration round: If the cutoff is set at the end of the first phase, FATB prioritizes sample A. If the cutoff is set at the end of the second phase, FATB prioritizes sample B. If the cutoff is set at the end of the third phase, FATB prioritizes sample B.

**Implementation:** FATB was implemented using PyTorch (Paszke et al. (2017)). Experiments on the CIFAR-10 and CIFAR-100 datasets were conducted on a server with 4 NVIDIA GeForce RTX 3090 GPUs, while experiments on the ImageNet-1k dataset were performed on a server with 4 NVIDIA GeForce RTX 5090 GPUs. For the CIFAR-10 and CIFAR-100 datasets, the ResNet-18 model was trained for 200 epochs with a batch size of 128. The optimizer used was SGD with a momentum of 0.9, weight decay of 5e-4, and a learning rate initialized at 0.1 following a cosine annealing schedule. For the ImageNet-1k dataset, the ResNet-50 model was trained for 90 epochs with a batch size of 256. The other hyperparameters (optimizer, momentum, weight decay, learning rate schedule) remained the same as those used for the CIFAR datasets.

### 4.2 RESULTS

The experimental results were generally obtained by evaluating outcomes at intervals of 10 epochs and comparing the results from the final epoch within each interval to determine the optimal cutoff epoch. However, under high pruning ratios, the result at the 10th epoch could underperform compared to random pruning. In such scenarios, the results from each epoch starting from the 2nd epoch within the first 10-epoch interval were compared instead.

**Benchmark Evaluation Results:** Table 1 records the results of training a ResNet-18 model on pruned subsets of the CIFAR-10 and CIFAR-100 datasets, selected by various data pruning algorithms. FATB generally outperforms the other benchmark algorithms. The only exception occurs on the CIFAR-10 dataset at a pruning ratio of pr = 0.7, where FATB's performance is slightly lower than the Forgetting algorithm but still superior to the others. Notably, FATB exhibits marked superiority under high pruning ratios. For instance, when training on CIFAR-100 with pr = 0.9, FATB achieves a performance 7.39 percent higher than Random Pruning, representing a 20 percent performance improvement relative to it, and outperforms the TDDS algorithm by 1.96 percent, constituting a 4.7 percent performance gain. These results demonstrate the outstanding effectiveness of FATB, its substantial performance gains, and its excellence as a data pruning method.

**Model Generalization Evaluation Results:** Tables 2 and 3 present the results of training the pruned subsets of CIFAR-10 and CIFAR-100 datasets, selected by various data pruning algorithms using a

Table 1: Comparison of the Top-1 accuracy achieved by a ResNet-18 model trained on the optimal data subsets for CIFAR-10 and CIFAR-100, obtained using different data pruning algorithms. The best results are highlighted in bold.

| Dataset | CIFAR-10 | | | | | CIFAR-100 | | | | |
|---|---|---|---|---|---|---|---|---|---|---|
| Pruning ratio | 0.3 | 0.5 | 0.7 | 0.8 | 0.9 | 0.3 | 0.5 | 0.7 | 0.8 | 0.9 |
| FATB | 95.61 | **95.49** | **93.73** | **91.15** | **82.9** | **78.17** | **75.27** | **67.65** | **60.24** | **43.23** |
| Static Random | 94.52 | 93.41 | 90.72 | 87.13 | 79.68 | 75.05 | 71.51 | 64.79 | 56.5 | 35.84 |
| TDDS | 95.27 | 95.19 | 92.93 | 89.62 | 80.49 | 77.84 | 74.79 | 66.87 | 59.92 | 41.27 |
| Moso | 95.03 | 94.25 | 91.06 | 85.83 | 71.69 | 73.99 | 70.36 | 61.27 | 49.47 | 30.23 |
| Forgetting | **95.83** | 95.38 | 86.61 | 61.51 | 43.91 | 75.41 | 65.54 | 49.44 | 35.41 | 19.32 |
| EZLN-20 | 95.1 | 94.3 | 82.7 | 57.3 | 38 | 72.32 | 63 | 41.5 | 29.7 | 19.1 |
| Whole Dataset | | | 95.67 | | | | | 78.66 | | |

Table 2: Comparative Top-1 accuracy results of a VGG-16 model trained on the optimal data subsets for CIFAR-10 and CIFAR-100, which were obtained using different data pruning algorithms with a ResNet-18 model. The best results are highlighted in bold.

| ResNet-18 → VGG-16 | | | | | | | | | | |
|---|---|---|---|---|---|---|---|---|---|---|
| Dataset | CIFAR-10 | | | | | CIFAR-100 | | | | |
| Pruning ratio | 0.3 | 0.5 | 0.7 | 0.8 | 0.9 | 0.3 | 0.5 | 0.7 | 0.8 | 0.9 |
| FATB | **94.22** | **93.55** | **91.3** | **87.65** | **78.8** | **73.93** | **70.47** | **62.96** | 55.48 | 38.48 |
| Static Random | 92.81 | 91.33 | 87.85 | 85.6 | 77.36 | 71.07 | 66.89 | 59.63 | 52.71 | 32.75 |
| TDDS | 94.1 | 93.32 | 90.53 | 87.23 | 78.34 | 73.55 | 69.97 | 62.93 | **56.04** | **42.1** |
| Whole Dataset | | | 95.67 | | | | | 78.66 | | |

ResNet-18 model, with VGG-16 and ShuffleNet-v2 models, respectively. FATB consistently outperformed random pruning across all scenarios and surpassed the TDDS algorithm in most scenarios. A notable result is that when training on CIFAR-100 with a pruning ratio of pr = 0.9, FATB achieved a performance 7.39 percent higher than Random Pruning, representing a 20 percent relative performance improvement, and outperformed TDDS by 1.96 percent, constituting a 4.7 percent relative gain. These results indicate that FATB possesses strong model generalization capabilities and is not limited to a specific model architecture.

**Large-Scale Dataset Evaluation Results:** Table 4 records the results on the ImageNet-1K dataset. FATB consistently outperformed all other algorithms, particularly at higher pruning ratios. For instance, at pr = 0.9, FATB's performance was 3.5 percent higher than Random Pruning and 1.9 percent higher than TDDS. This confirms the algorithm's excellent effectiveness on large-scale datasets, making it suitable for selecting core subsets from massive datasets.

Table 3: Comparative Top-1 accuracy results of a ShuffleNet-v2 model trained on the optimal data subsets for CIFAR-10 and CIFAR-100, which were obtained using different data pruning algorithms with a ResNet-18 model. The best results are highlighted in bold.

| ResNet-18 → ShuffleNet-v2 | | | | | | | | | | |
|---|---|---|---|---|---|---|---|---|---|---|
| Dataset | CIFAR-10 | | | | | CIFAR-100 | | | | |
| Pruning ratio | 0.3 | 0.5 | 0.7 | 0.8 | 0.9 | 0.3 | 0.5 | 0.7 | 0.8 | 0.9 |
| FATB | 92.28 | 92.07 | **89.26** | **85.78** | **73.49** | **70.22** | **66.76** | **60.3** | **53.2** | **39.92** |
| Static Random | 90.69 | 89.56 | 86.92 | 83.93 | 70.29 | 68.13 | 64.59 | 57.15 | 50.09 | 33.35 |
| TDDS | **92.81** | **92.69** | 87.82 | 79.13 | 64.51 | 69.45 | 66.39 | 58.9 | 51.62 | 34.13 |
| Whole Dataset | | | 95.67 | | | | | 78.66 | | |

Table 4: Comparison of the Top-1 accuracy achieved by a ResNet-50 model trained on the optimal data subset for ImageNet-1K, obtained using different data pruning algorithms. The best results are highlighted in bold. The best results are highlighted in bold.

| Dataset | ImageNet-1K | | |
|---|---|---|---|
| Pruning ratio | 0.7 | 0.8 | 0.9 |
| FATB | **67.504** | **63.516** | **56.418** |
| Static Random | 66.874 | 62.926 | 52.876 |
| TDDS | 67.206 | 62.984 | 54.51 |
| Whole Dataset | 72.972 | | |

Table 5: For the four basic scenarios, the optimal data subsets of the CIFAR-10 and CIFAR-100 datasets are obtained using a ResNet-18 model. Subsequently, the comparative Top-1 accuracy results are evaluated by training a ResNet-18 model on these subsets. Each result is presented in two parts: the top value indicates the accuracy, and the bottom value denotes the cutoff iteration number. The highest accuracy for each case is highlighted in bold, and if the corresponding cutoff iteration number for the highest accuracy is also the best, it is bolded as well.

| Dataset | CIFAR-10 | | | | | CIFAR-100 | | | | |
|---|---|---|---|---|---|---|---|---|---|---|
| Pruning ratio | 0.3 | 0.5 | 0.7 | 0.8 | 0.9 | 0.3 | 0.5 | 0.7 | 0.8 | 0.9 |
| BSA | 95.61 | 95.49 | 93.73 | **91.15** | **82.9** | **78.17** | **75.27** | **67.65** | **60.24** | **43.23** |
| | (189) | (159) | (119) | **(59)** | **(9)** | **(99)** | **(79)** | **(19)** | **(5)** | **(3)** |
| BSB | 95.65 | **95.56** | **93.8** | 90.35 | 78.36 | 78.03 | 75.16 | 65.86 | 54.39 | 30.55 |
| | (169) | (179) | **(119)** | (69) | (29) | (189) | (89) | (29) | (9) | (7) |
| BSC | **95.67** | 95.47 | 92.74 | 68.66 | 32.6 | 77.78 | 69.79 | 43.23 | 46.51 | 17.98 |
| | **(99)** | (89) | **(179)** | (199) | (109) | (189) | (189) | (9) | (1) | (1) |
| BSD | 91.44 | 88.55 | 83.93 | 80.93 | 69.87 | 74.8 | 70.27 | 62.78 | 57.03 | 43.18 |
| | (189) | (199) | (199) | (199) | (189) | (179) | (199) | (179) | (199) | (179) |
| Whole Dataset | 95.67 | | | | | 78.66 | | | | |

## 4.3 ANALYSIS OF THE FOUR BASIC SCENARIOS

For the four basic scenarios, proxy model training was first conducted concurrently to obtain their respective statistical results. Subsequently, data pruning was performed based on these results to derive core data subsets, and finally, models were trained separately to obtain their respective performance outcomes (as shown in Table 5).

Experimental results indicate that the core data subset selected based on basic scenario (a) yielded better training outcomes in most scenarios compared to the other three scenarios. Additionally, its corresponding cutoff iteration number was superior to those of the other basic scenarios. At low pruning rates, the results of scenario (a) were slightly lower than those of scenarios (b) and (c), but the difference was negligible.

In summary, basic scenario (a) is the primary factor influencing the selection results of the core data subset.

## 4.4 HYPERPARAMETER ANALYSIS

To determine the appropriate cutoff iteration round for FATB, experimental results under different rounds are provided (see Figures 3 and 4), with the training accuracy of random data pruning algorithms included as a baseline. The results indicate that as the pruning ratio increases, FATB allows earlier termination of surrogate model training, thereby reducing computational cost. Moreover, a consistent trend is observed: on clean datasets, a smaller pruning ratio requires a later cutoff round, while a larger ratio corresponds to an earlier cutoff. Figures 3 and 4 clearly show that without a

predefined cutoff—i.e., using the final iteration—FATB performs poorly, even significantly worse than random pruning algorithms, as evidenced by the considerable accuracy gaps on CIFAR-10 and CIFAR-100. In clean data environments, effective subsets must include early hard-to-learn samples to enhance model discriminability, necessitating an earlier cutoff to improve subset informativeness.

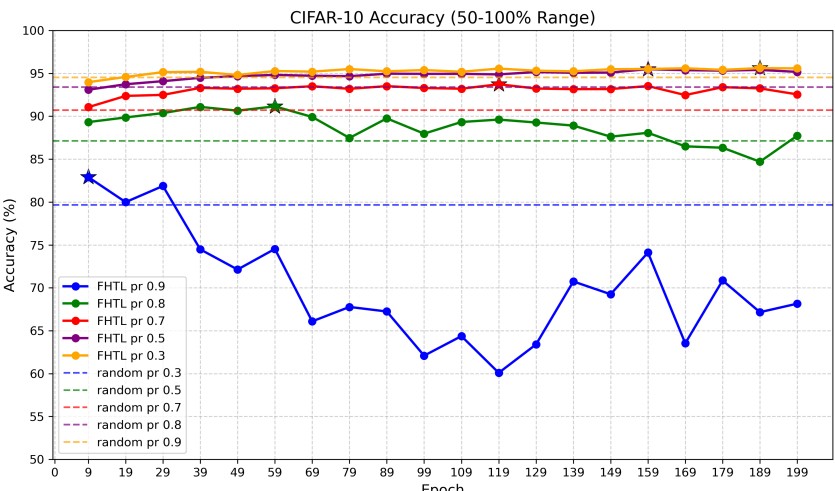

Figure 3: FATB obtains the optimal data subset for the CIFAR-10 dataset via surrogate training of a ResNet-18 model at different cutoff iteration rounds. A comparative analysis of the Top-1 accuracy (Acc) achieved by training the ResNet-18 model on these respective optimal subsets is presented. The star symbol (★) denotes the highest accuracy attained for the corresponding pruning ratio.

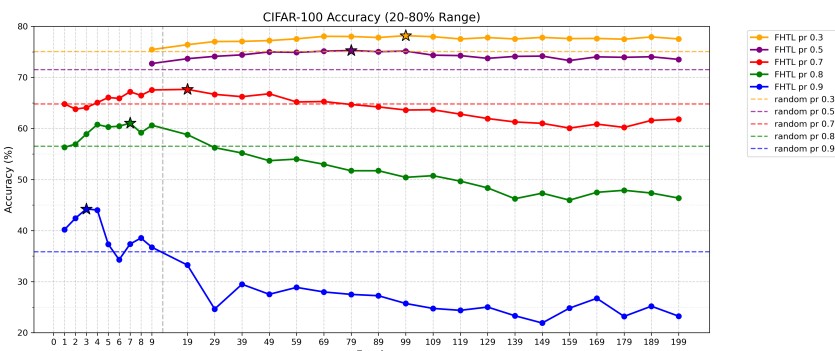

Figure 4: FATB obtains the optimal data subset for the CIFAR-100 dataset via surrogate training of a ResNet-18 model at different cutoff iteration rounds. A comparative analysis of the Top-1 accuracy (Acc) achieved by training the ResNet-18 model on these respective optimal subsets is presented. The star symbol (★) denotes the highest accuracy attained for the corresponding pruning ratio.

## 5 CONCLUSION

This paper proposes FATB, an efficient zeroth-order data pruning algorithm that identifies redundant and noisy samples solely based on loss variation during surrogate model training. By avoiding gradient and Hessian computations, it significantly reduces computational cost. FATB addresses the limitations of existing methods at high pruning ratios through early training termination. However, as it prioritizes training utility over sample content, its ability to detect corrupted or low-quality data remains limited. Future work will integrate content-based analysis to improve core subset selection and enhance model performance.

ACKNOWLEDGMENTS

The views, opinions, and/or findings contained in this article are those of the authors.

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

## A    Noisy Data Evaluation

**Noisy Data Evaluation Results:** Label Noise: Table 6 records the results on CIFAR-10 and CIFAR-100 datasets with partially corrupted labels. FATB consistently outperformed all other baselines and significantly improved model performance, demonstrating that the selected subsets contained less label-noisy data. Consequently, this algorithm exhibits a strong capability for identifying label noise and could, to some extent, be specialized for label noise cleaning tasks. Sample Corruption Noise: Table 7 records the results on CIFAR-10 and CIFAR-100 datasets modified by a series of sample corruption methods. FATB consistently outperformed random pruning and surpassed TDDS in some scenarios, with only minor performance gaps where it slightly trailed TDDS. This suggests the algorithm can identify certain sample corruption noise to a degree but still has room for improvement.

## B    Algorithm Implementation and Hyperparameter Details

### B.1    Algorithm Implementation:

---

**Algorithm 1** Transition From Above-Average To Below-Average (FATB)

---

**Input:** Training dataset $S = \{(x_i, y_i)\}_{i=1}^n$; Total iterations $T$; Cutoff iteration $DT \in [0, T]$; Pruning ratio $pr \in [0, 1]$
**Output:** Pruned dataset $Z \subset S$ with $|Z| = n \times (1 - pr)$
  1: Initialize sample transition count array $C[1..n] \leftarrow 0$
  2: Initialize previous high-loss sample set $P \leftarrow \emptyset$
  3: **for** $t = 0$ **to** $T - 1$ **do**
  4:     According to Equation 2, compute average loss $L_t$
  5:     According to Equation 3, identify current high-loss samples $H_t$
  6:     **if** $t > 0$ **then**
  7:         According to Equation 10, find transition samples $T_t$
  8:         **for** each sample $i \in T_t$ **do**
  9:             $C[i] \leftarrow C[i] + 1$ {Increment transition count}
10:         **end for**
11:     **end if**
12:     Update previous set: $P \leftarrow H_t$
13:     **if** $t \geq DT$ **then**
14:         Sort samples by descending transition count $C$
15:         Select top $n \times (1 - pr)$ samples with highest counts
16:         $Z \leftarrow$ selected samples
17:         **return** $Z$
18:     **end if**
19: **end for**
20: **return** $Z$

---

### B.2    Hyperparameter Details:

The specific hyperparameters used by FATB in the mentioned tables are listed in the format (pruning ratio, cutoff iteration round) as follows: Table 1: CIFAR-10: (0.3, 189), (0.5, 159), (0.7, 119), (0.8, 59), and (0.9, 9); CIFAR-100: (0.3, 99), (0.5, 79), (0.7, 19), (0.8, 7), and (0.9, 3). Tables 2, 3, and 7 are consistent with Table 1. Table 6: CIFAR-10: (0.3, 109), with the remaining parameters consistent with Table 1. Table 4: ImageNet-1K: (0.7, 89), (0.8, 89), and (0.9, 49).

Table 6: First, introduce label noise by randomly modifying 20 percent of the sample labels in the CIFAR-10 and CIFAR-100 datasets, respectively, such that these labels are inconsistent with the original ones. Subsequently, various data pruning algorithms are employed to obtain the optimal data subsets from the modified datasets using a ResNet-18 model. The comparative results of Top-1 accuracy are then evaluated by training a ResNet-18 model on these subsets. The best results are highlighted in bold.

| Label Noise | | | | | | | | | | |
|---|---|---|---|---|---|---|---|---|---|---|
| Dataset | CIFAR-10 | | | | | CIFAR-100 | | | | |
| Pruning ratio | 0.3 | 0.5 | 0.7 | 0.8 | 0.9 | 0.3 | 0.5 | 0.7 | 0.8 | 0.9 |
| FATB | **91.3** | **90.48** | **89.59** | **79.79** | **71.79** | **71.9** | **71.0** | **62.75** | **53.79** | **35.7** |
| Static Random | 84.75 | 81.7 | 77.94 | 72.52 | 61.59 | 62.03 | 58.38 | 50.22 | 39.61 | 23.95 |
| TDDS | 86.05 | 85.43 | 81.94 | 77.43 | 62.89 | 65.92 | 63.19 | 54.48 | 44.9 | 28.4 |
| Whole Dataset | 88.28 | | | | | 64.77 | | | | |

Table 7: First, introduce image corruption by applying five distinct methods—Gaussian noise, random occlusion, resolution degradation, fog simulation, and motion blur—to randomly and non-overlappingly corrupt 4 percent of the samples each in the CIFAR-10 and CIFAR-100 datasets, resulting in a total of 20 percent corrupted samples with perceptible alterations from their originals. Subsequently, various data pruning algorithms are employed to identify the optimal data subsets from the corrupted datasets using a ResNet-18 model. The comparative Top-1 accuracy results are then obtained by training and evaluating a ResNet-18 model on these pruned subsets. The best results are highlighted in bold.

| Image Corruption | | | | | | | | | | |
|---|---|---|---|---|---|---|---|---|---|---|
| Dataset | CIFAR-10 | | | | | CIFAR-100 | | | | |
| Pruning ratio | 0.3 | 0.5 | 0.7 | 0.8 | 0.9 | 0.3 | 0.5 | 0.7 | 0.8 | 0.9 |
| FATB | **94.01** | **93.86** | 90.48 | 86.15 | **79.77** | 74.93 | 71.3 | 62.87 | 55.82 | 37.38 |
| Static Random | 93.01 | 91.29 | 88.15 | 84.67 | 72.66 | 71.2 | 67.42 | 59.79 | 49.55 | 30.99 |
| TDDS | 93.69 | 93.27 | **91.25** | **87.96** | 79.1 | **75.02** | **72.78** | **65.86** | **57.88** | **39.61** |
| Whole Dataset | 94.05 | | | | | 75.06 | | | | |