# OpenReview forum: "Data Pruning: Counting the Frequency of Loss Transition from Above-Average to Below-Average (FATB) During Early Training"
_ICLR.cc/2026/Conference — ICLR 2026 Conference Withdrawn Submission_

### Official Review · Reviewer_h5vm · 2025-10-25

**Soundness:** 3
**Presentation:** 2
**Contribution:** 3
**Rating:** 6
**Confidence:** 4

**Summary:**

This paper proposes FATB, a zero-order data pruning method that identifies informative samples by counting how often a sample’s loss transitions from being above the dataset’s average loss to below it during early training epochs of a proxy model. The core idea is that such transitions reflect a sample’s contribution to learning dynamics. FATB avoids expensive gradient or Hessian computations, relying only on per-sample and average loss values across consecutive iterations. An early stopping strategy is employed to determine an optimal cutoff iteration for counting transitions—earlier cutoffs are used at higher pruning ratios. The method is evaluated on CIFAR-10, CIFAR-100, and ImageNet-1K with various architectures (ResNet, VGG, ShuffleNet), showing consistent improvements over strong baselines like Forgetting, EL2N, MoSo, and TDDS—especially at high pruning ratios (e.g., 90%). The paper also includes ablation studies on the four possible loss-transition scenarios, confirming that the “above-to-below” transition (BSA) is the most predictive of sample utility.

**Strengths:**

**Simplicity and Efficiency:**

FATB is a truly zero-order method requiring only loss values, which making it highly scalable and computationally lightweight compared to first- or second-order pruning techniques. This is a significant practical advantage for large-scale settings.


**Strong Empirical Performance:**

The method consistently outperforms existing approaches across datasets, models, and pruning ratios, with notable gains at high pruning levels (e.g., +7.39% over random on CIFAR-100 at 90% pruning). This addresses a known weakness in prior methods that degrade sharply under aggressive pruning.

**Well-Motivated Core Insight:**

The focus on transient loss behavior (rather than static difficulty or persistent states) aligns with recent understanding of training dynamics (e.g., dataset cartography). The ablation in Table 5 convincingly shows BSA is superior to other transition types.

**Robustness and Generalization:**

Experiments demonstrate cross-architecture generalization (prune with ResNet, train with VGG/ShuffleNet) and resilience to label noise and image corruption, suggesting FATB selects not just easy samples but informative ones.

**Weaknesses:**

**Ambiguity in “Informational Value”:**

The term “informational value” is used intuitively but not formally defined. It’s unclear whether FATB prioritizes hard examples, diverse examples, or something else. Linking this to known data selection principles (e.g., diversity, difficulty, coverage) would clarify the mechanism.

**Computational Cost of Proxy Training:**

Although cheaper than gradient-based methods, FATB still requires training a proxy model for tens or hundreds of epochs. For extremely large datasets (e.g., web-scale), even this may be prohibitive. A discussion of wall-clock time or FLOPs vs. baselines would be valuable.


**Experiments**

We recommend that authors conduct experiments on larger-scale datasets.

**Questions:**

See Weaknesses

---

### Official Review · Reviewer_8Yzc · 2025-11-01

**Soundness:** 2
**Presentation:** 1
**Contribution:** 2
**Rating:** 2
**Confidence:** 4

**Summary:**

This paper introduces the FATB method, which identifies a core subset of data by comparing the loss trajectories of individual samples with the average loss across all training samples. In particular, the authors observe that samples whose losses frequently transition from above-average (hard-to-learn) to below-average (easy-to-learn) during training tend to be highly informative and should therefore be retained. Using this criterion, the proposed method achieves state-of-the-art performance on CIFAR-10, CIFAR-100, and ImageNet across most pruning ratios. Moreover, the appendix includes experiments on noisy datasets with corrupted labels and images, demonstrating that the method remains effective under such conditions.

**Strengths:**

1. The Introduction and Related Work sections are well-written and comprehensive. Prior works are thoroughly reviewed across static and dynamic data pruning methods.
2. The proposed method, FATB, is novel in that it compares each sample's loss trajectory with the averaged loss trajectory. This approach captures temporal information during training while implicitly reflecting dataset difficulty, since individual trajectories are compared against the average. The method achieves state-of-the-art performance across various datasets, including CIFAR-10, CIFAR-100, and ImageNet. Additionally, the authors tested the method on noisy scenarios and demonstrate that it can successfully discard noisy samples, achieving good test performance even compared to full-training.

**Weaknesses:**

**Major**

1. The abstract should be rewritten for clarity. The current version contains many undefined terms and overly abstract wording. For example:
    1. The naming of FATB is not explained.
    2. In line 20, the authors mention comparing the loss value of each sample with the average loss across all samples, resulting in *four fundamental scenarios*. However, there is no detailed explanation of what these four scenarios actually are. Since the abstract keeps referring to them, readers may find it confusing. Consider adding a pointer to Figure 1 at least.
    3. Moreover, since the method focuses only on one transition, which is “*from above-average to below-average loss”.*  there is no need to mention *four scenarios*. It would be clearer to describe only the relevant transition and emphasize the core idea (lines 26–28): “the loss of a single sample transitions ~ to estimate the informational value of the sample.” This sentence best captures what FATB does, while lines 23–25 can be omitted as they do not contribute meaningfully to describing the algorithm.
    4. In line 31, the phrase “Extensive experiments” seems like an overclaim, as the presented experiments are not that comprehensive.
2. Section 3 lacks the clarity necessary for comprehension. The main issues are as follows:
    1. The function $\ell(\cdot, \cdot)$ is not defined. At minimum, the authors should state that it represents the loss function.
    2. In line 144, the authors define $U = {(x_m, y_m)}\_{m=1}^M$ as the core data subset. However, since the training dataset $S$ was defined as ${(x_n, y_n)}_{n=1}^N$, this definition implies that the core subset simply consists of the first $M$ samples in $S$, without any selection process. The authors should redefine $U$.
    3. It is recommended to denote the pruning ratio with a mathematical symbol (e.g., $r$) rather than plain text (pr) for consistency in notation.
    4. In Section 3.2, the term FATB never appears, even though “Method” is the title of the section. The authors should explicitly state that “This method is referred to as FATB.”
    5. In Section 3.1, vectors are written using bold notation (e.g., $\mathbf{x}$), but starting from Section 3.2, the boldface is dropped (e.g., $x$). For consistency, the authors should choose one notation style and use it throughout.
    6. Method section (lines 189–249) needs a substantial rewrite.
        - Over-definition without use. Equations (5)–(10) introduce several constructs that are never used later. If they are not required by the final algorithm, they should be removed or moved to the appendix.
        - Focus the method on the single relevant transition. The core idea of FATB is the transition from *above-average* to *below-average* loss. Other “scenarios” are ancillary and better treated as ablations, not as main-text definitions. Defining scenario (a), scenario (b), … in the body then discarding them obscures the contribution.
        - Reduce indicator-function nesting. The manuscript repeatedly defines events via nested indicator functions (e.g., $\mathrm{STP}(x_n,y_n;t)=\mathbf{1}({\mathrm{BSA}})$ where $\mathrm{BSA}=\mathbf{1}({\mathrm{HTL}\land \mathrm{ETL}})$, and so on). This stacking is hard to read and easy to misinterpret. Define events once using plain-language predicates (or simple inequalities), then use a single indicator if needed.
    7. The key quantity, FATB for each $(x_n, y_n)$, is defined as $\sum_{t=1}^T \mathrm{STP}((x_n, y_n); t)$, but it appears only as a sentence within the “In Summary” paragraph. This equation should be explicitly displayed in the Method section, preceded by a clear explanation that $\mathrm{BSA}$ is a per-time event and that FATB simply counts how many times it occurs over $t = 1, \dots, T$, where $T$ denotes the cutoff iteration.
    8. In addition, consider adding a brief interpretation of FATB (for example, “higher values indicate more frequent transitions from hard to easy, implying greater informational value and higher priority for retention”).
3. Rationale for selecting BSA is unclear. Additional ablations on the other three scenarios are needed. Although Table 5 reports results for the alternatives, accuracy alone does not justify the choice. Please explain why BSA should be considered more informative than the other scenarios, and include further ablations beyond accuracy.
4. It appears that only a single random seed was used in the experiments. Since there is no mention of the number of runs or any reported standard deviation, the results seem to be based on a single trial. This raises concerns about statistical reliability. The authors should rerun each experiment with at least five different random seeds and report the mean and standard deviation. Without such analysis, the current results cannot be considered trustworthy.
5. The set of baselines compared in the paper is outdated. To properly situate the method's performance within the current literature, it is crucial to include comparisons against recent and relevant SOTA methods. A few examples include CCS [1], D2-pruning [2], Dyn-Unc [3], and DUAL (with $\beta$ sampling) [4].
6. Although the method has only one hyperparameter, the cutoff iteration, tuning this parameter is essential and incurs a substantial computational cost. In particular, the authors search over all possible cutoff iterations every 10 epochs for each pruning ratio, which requires full training runs. Therefore, the claim that the method is computationally efficient because it can terminate early is not convincing. On the contrary, since the cutoff iteration must be searched for every pruning ratio, the method is computationally inefficient. Moreover, if a new pruning ratio (for example, 40%) is introduced, the search process must be repeated, further increasing the computational burden.
7. Algorithm 1 in Appendix B should be rewritten for clarity.
    - Avoid using plain text within the pseudocode. For example, lines such as “According to Equation 2, compute average loss $L_t$,” “Select top $n \times (1-pr)$ samples with highest counts,” and “$Z \gets \text{selected samples}$” should be replaced with concise, formal pseudocode expressions.
    - If the cutoff iteration $DT = T$ (the total number of iterations), the algorithm cannot return the selected samples, because $t$ runs only from 0 to $T-1$, and the if condition is satisfied only when $t \geq DT$. This logic should be adjusted.
    - When the if condition is not satisfied, the algorithm still returns $Z$ (line 20), but $Z$ has never been defined in that case. Moreover, there is no need to return $Z$ if the if block is never executed.
    - For clarity, change the if condition to $t = DT$.
    - Do not use set notation ${}$ as a comment.
    - The variable $P$ (line 12) is introduced but never used anywhere. Clarify its purpose or remove it.

    In addition to these points, many other lines contain similar issues, so a full rewrite of the algorithm is recommended.


**Minor**

1. The explanation of the results is overly verbose. The section mainly lists percentage improvements over Random or TDDS for every experiment, which does not provide meaningful insight. Summaries should focus on key trends or qualitative interpretations rather than raw percentage comparisons.
2. On page 7, consider swapping the positions of Table 3 and the Large-Scale Dataset Evaluation Results paragraph to improve readability and logical flow.
3. Comments on Figures and Tables:
    1. The width of several tables exceeds the text area. Please resize them to fit within the main text width.
    2. Figure labels and legends are too small and difficult to read; increase their font size for clarity.
    3. The term “Static Random” seems unnecessary. Since all compared methods are static pruning methods and no Dynamic Random baseline is presented, this label can be simplified to “Random”.
    4. In Tables 6 and 7 (Appendix), descriptions such as experiment setting should appear in the main text, not in the caption. Captions should contain only information essential for interpreting the table or figure.
    5. In Table 5, all epoch numbers end with 9. It would be clearer to start counting epochs from 1 and align the values to round numbers rather than zero-based indexing.
    6. Only the ImageNet results are formatted with three decimal places, while others use two. Please standardize the formatting for consistency.
4. Please include a pointer in the main text to the results on the noisy dataset. Although these results are presented in the Appendix, readers may miss them without any reference in the main body.
5. Throughout the paper, replace zero-order with zeroth-order for consistent and correct terminology.
6. In line 230, change table 5 to Table 5 for consistency with capitalization style.
7. In Table 1, correct EZLN to EL2N.
8. Change the section title Proposed Methods to Proposed Method since the authors only propose FATB.
9. The conclusion introduces new information. This section should focus on summarizing results and suggesting future work. The content in lines 484–485 should be explained earlier in the main text instead.
10. In Appendix A, apply boldface formatting to Label Noise and Sample Corruption Noise.

---

**References:**

[1] Coverage-centric Coreset Selection for High Pruning Rates, ICLR 2023.

[2] D2 Pruning: Message Passing for Balancing Diversity and Difficulty in Data Pruning, ICLR 2024.

[3] Large-scale Dataset Pruning with Dynamic Uncertainty, CVPR Workshop 2024.

[4] Lightweight Dataset Pruning without Full Training via Example Difficulty and Prediction Uncertainty, ICML 2025.

**Questions:**

1. There are no results for ImageNet with low pruning ratios. Please include them or explain why they are omitted.
2. The paper should report how many noisy samples were actually removed. Since the authors claim that their method effectively filters out noisy data, results should include quantitative evidence beyond accuracy, such as the proportion of noisy samples eliminated.
3. A visual analysis would be helpful to illustrate which samples are removed and which are retained by the method.
4. Please clarify what each point represents in Figure 2.
5. Instead of using Sample A and Sample B, use consistent mathematical notation such as $\mathbf{x}_i$ and $\mathbf{x}_j$.
6. The phrasing in Section 4.4 seems repetitive; consider revising to avoid redundancy.

---

### Official Review · Reviewer_Vrhe · 2025-11-01

**Soundness:** 3
**Presentation:** 2
**Contribution:** 3
**Rating:** 6
**Confidence:** 4

**Summary:**

This paper proposes a novel zero-order data pruning algorithm named FATB (Frequency of loss transition from Above-average to Below-average), which identifies a core subset of informative samples by counting how often a sample’s loss transitions from above the dataset average to below it during early training epochs. The method avoids expensive gradient or Hessian computations and leverages an early stopping strategy to select high-value samples, especially effective under high pruning ratios. Experiments on CIFAR-10, CIFAR-100, and ImageNet-1K with various architectures (ResNet, VGG, ShuffleNet) show that FATB consistently outperforms existing static pruning baselines, including Forgetting, EL2N, MoSo, and TDDS, particularly when pruning ratios exceed 70%. The paper also includes ablation studies validating the choice of the core transition scenario (BSA) and demonstrates robustness to label noise and image corruption.

**Strengths:**

1. Conceptual Simplicity and Efficiency: FATB is a zero-order method that relies only on per-sample and average loss comparisons across consecutive epochs, making it computationally lightweight and easy to implement.

2. Strong Empirical Performance: The method shows consistent gains over strong baselines across multiple datasets and model architectures, especially at high pruning ratios where many prior methods degrade sharply.

3. Well-Motivated Core Scenario: The focus on the “above-to-below average loss” transition is theoretically grounded in training dynamics literature and empirically validated through ablation (Table 5).

4. Practical Early-Stopping Strategy: The adaptive cutoff iteration (earlier for higher pruning ratios) is both intuitive and effective, addressing a known weakness of static pruning at aggressive compression levels.

5. Robustness to Noise: FATB shows promising results on datasets with label noise and image corruption, suggesting it can implicitly filter out low-quality samples.

**Weaknesses:**

1. Limited Experimental Scale: While ImageNet-1K is included, the largest model used is ResNet-50, and experiments do not extend to modern large-scale vision models (e.g., ViTs) or truly massive datasets (e.g., full-scale LAION or JFT). The method’s scalability to billion-sample regimes remains unverified.

2. Early Stopping Heuristic: The cutoff iteration is selected via validation over epochs, which requires multiple full proxy trainings or at least epoch-wise evaluation, a cost not fully accounted for in the “efficiency” claim. An automated or theoretically justified stopping criterion would strengthen the method.

3. No Analysis on Data Distribution Shift: It is unclear how FATB performs when the pruned subset is used for transfer learning or under domain shift, which is critical for real-world deployment.

**Questions:**

See Weakness

---

### Official Review · Reviewer_Emq6 · 2025-11-04

**Soundness:** 3
**Presentation:** 2
**Contribution:** 2
**Rating:** 2
**Confidence:** 4

**Summary:**

The paper introduces FATB (Frequency of Loss Transition from Above-average to Below-average), a novel, zero-order data pruning algorithm. The core idea is to identify the most informative samples in a dataset by analyzing their loss dynamics during the early stages of model training. It specifically counts the number of times a sample's loss transitions from being above the average in one epoch to below the average in the next.

**Strengths:**

- The proposed FATB metric is intuitive and simple to implement. Framing the data pruning problem around the dynamics of a sample's loss relative to the average is a creative and effective approach.
-  As a zero-order algorithm, FATB is significantly cheaper than first-order (gradient-based) or second-order methods.

**Weaknesses:**

1. Hyperparameter Sensitivity (The Cutoff Iteration): The algorithm's performance is critically dependent on the cutoff iteration (DT). As shown in Figures 3 and 4, choosing a suboptimal cutoff can lead to performance that is even worse than random pruning. The paper demonstrates a clear trend (higher pruning ratio correlates with an earlier optimal cutoff), but it doesn't propose a practical, low-cost method for determining this hyperparameter for a new dataset or model.
2.  The paper provides strong empirical evidence for why the FATB (BSA) scenario works best, but no theoretical motivation. A more formal explanation would elevate the work beyond a purely empirical finding.
3. Several ablations or clarifications are needed to prove the effectiveness of the method (see question 3, 4, 5, 6).

**Questions:**

1. For TDDS performance, there is a discrepancy between the reported one and reported in the TDDS original paper. For example, for CIFAR-10 70% pruning rate, the acc is 93.92 ±0.04 in the TDDS paper, which is higher (or similar) than the reported FATB  93.73. The whole dataset performance in CIFAR-10 is similar in the two papers.
2. Still not clear why the late cutoff would cause a worse performance even than random? Before the cutoff epoch, the model is trained with the whole dataset?
3. Whether the mean loss of the whole dataset is optimal? It ignores class imbalance and per-class difficulty.  It also ignores the effect of data augmentation (e.g., different random crops).
4. Missing ablation on the averaging window: Only adjacent-epoch transitions are counted. Would a smoothed mean or k-epoch trend improve robustness?
5. The method counts events but ignores the margin by which loss crosses the mean. A small, noisy crossing counts the same as a large one.
6. The BSA sounds contrary to forgetting events. The BSA keeps those losses from high to low while the forgetting events keep that accuracy from high to low (i.e, loss from low to high). This needs to be claimed more clearly.

---

### Official Review · Reviewer_62iR · 2025-11-07

**Soundness:** 1
**Presentation:** 1
**Contribution:** 1
**Rating:** 0
**Confidence:** 5

**Summary:**

The paper addresses data pruning during model training. Specifically, the approach attempts to remove redundant and noisy samples.
The main idea of the work is to compare the sample loss with the mean loss of all samples and use that to determine the quality of the traning samples.

**Strengths:**

Only strength of the paper is that it atempts to address a very important problem.

**Weaknesses:**

The paper has lot of weaknesses.

1. The main idea on which the work is standing is very naive, and oversimplifies the hardness of samples.

2. It has no theoretical basis and the idea seems bit too adhoc. No proof or justification is given, why the conditions should be true.

3. Experimental setup is very weak. Choice of baselines are poor. The results too are equally poor and nowhere close to the expected level.

**Questions:**

No additional question.

---

### Note · Authors · 2025-11-15

I have read and agree with the venue's withdrawal policy on behalf of myself and my co-authors.